# Synthesis of Starch-Based Ag_2_[Fe (CN)_5_NO] Nanoparticles for Utilization in Antibacterial and Wound-Dressing Applications

**DOI:** 10.3390/antiox13020154

**Published:** 2024-01-25

**Authors:** Yuting Lu, Anbazhagan Sathiyaseelan, Xin Zhang, Lina Zhang, Kiseok Han, Myeong Hyeon Wang

**Affiliations:** 1Department of Bio-Health Convergence, Kangwon National University, Chuncheon 24341, Republic of Korea; yutinglu0614@gmail.com (Y.L.); sathiyaseelan.bio@gmail.com (A.S.); zhangxin199708@gmail.com (X.Z.); 1911880536z@gmail.com (L.Z.); seq0120@gmail.com (K.H.); 2KIIT (Kangwon Institute of Inclusive Technology), Kangwon National University, Chuncheon 24341, Republic of Korea

**Keywords:** starch, silver nitroprusside nanoparticles, antibacterial, wound healing, cytotoxicity

## Abstract

Bacterial infections can lead to the formation of chronic wounds and delay the wound-healing process. Therefore, it is important to explore safe and efficient antimicrobial agents that have wound-healing and biocompatible properties. In this study, novel starch-fabricated silver nitroprusside nanoparticles (S-AgNP NPs) were prepared for biocompatible wound-healing applications. The study showed that S-AgNP NPs are spherical, with an average size of 356 ± 22.28 d. nm and zeta potential of −27.8 ± 2.80 mV, respectively. Furthermore, the FTIR and XRD results showed that S-AgNP NPs have functional groups and crystal structures from the silver nitroprusside nanoparticles (AgNP NPs) and starch. Additionally, S-AgNP NPs showed excellent bacterial and biofilm inhibition on *B. cereus* (15.6 μg/mL), *L. monocytogenes* (15.6 μg/mL), *S. aureus* (31.3 μg/mL), *E. coli* (31.3 μg/mL) and *S. enterica* (62.5 μg/mL). Moreover, S-AgNP NPs promoted cell migration and proliferation at a concentration of 62.5 μg/mL compared to AgNP NPs. Meanwhile, S-AgNP NPs had good biocompatibility and low cytotoxicity compared to AgNP NPs. Therefore, this study provided new ideas for the development of wound-healing agents with bacteriostatic properties in chronic wounds.

## 1. Introduction

The skin is an important part of the human body. It regulates body temperature, protects the body from pathogens and external stimuli. The skin is easily damaged by physical or chemical injuries, leading to the appearance of wounds [1]. Wound healing and repair are a long and complex process. Wounds can be affected by a variety of factors at any stage, such as inflammation, extensive burns, bacterial infections, and diabetes [2]. Wounds are classified into acute and chronic wounds. Acute wounds are capable of eventually restoring structural integrity within a certain period [3]. However, chronic wounds are difficult to treat and are accompanied by wound infections and inflammation. This eventually leads to tissue necrosis. Bacteria play a major role in wound infection and form biofilms on the surface or inside chronic wounds, preventing healing. Bacteria secrete tissue-destroying enzymes, phagocytosis-resistant and adhesion-resistant exotoxins, and endotoxins in large quantities, resulting in the formation of chronic wounds [4]. The presence of drug-resistant bacteria can greatly reduce the efficacy of commonly used antibiotics [5]. Bacterial biofilms also stimulate neutrophils and pro-inflammatory macrophages to fight against the body’s immune system, which promotes the accumulation of inflammatory factors and exacerbates inflammation [6].

Nanoparticles are highly interesting because of their nano-size and high specific surface area, which allow them to penetrate cell membranes and act directly on pathogens. For instance, silver nanoparticles (AgNPs) have broad-spectrum antimicrobial capabilities that can act against a wide range of drug-resistant bacteria, and they have great potential for applications such as wound dressings and bacteriostatic agents [7]. In addition to this, AgNPs can be used for gene delivery, altering gene expression, and protein synthesis associated with the wound-healing process [8]. Nanoparticles also have an effect on collagen deposition and alignment during the wound-healing process, accelerating wound healing [9].

However, AgNP NPs are novel and formed by mixing and reducing silver nitrate with sodium nitroprusside. Sodium nitroprusside, a member of the prussides family, consists of an iron nucleus, multiple cyanide ion molecules, and one molecule of the nitroso ion [10]. Sodium nitroprusside has a wide range of pharmaceutical applications. Because of its ability to release NO, sodium nitroprusside can dilate blood vessels and regulate wound healing [11]. In recent years, AgNP NPs have shown potential for medical applications due to their excellent physicochemical properties, antimicrobial properties, and biocompatibility [12]. AgNP NPs have a strong inhibitory effect on both Gram-negative and Gram-positive bacteria and promote macrophage polarization to accelerate wound healing in mice [13,14]. In addition, AgNP NPs can be combined with hydrophobic cotton to be used as wound dressings [12]. However, AgNP NPs also face the problem of high toxicity in the human body due to the high silver content.

Therefore, this study aimed to combine starch with AgNP NPs to synthesize S-AgNP NPs. Furthermore, the physical and physiological properties of S-AgNP NPs were analyzed by analytical techniques to explore the application of S-AgNP NPs in wound healing and antibacterial applications. Moreover, the cytotoxicity and biocompatibility of S-AgNP NPs were further investigated by examining normal cells and red blood cells.

## 2. Materials and Methods

The pathogens, including *Bacillus cereus* (ATCC 14579), *Listeria monocytogenes* (ATCC 15313), *Staphylococcus aureus* (ATCC 19095), *Escherichia coli* (ATCC 43888), and *Salmonella typhi enterica* (ATCC14028), were obtained from the American Type Culture Collection (ATCC). Silver nitrate, sodium nitroprusside, and starch were purchased from Sigma Aldrich (St. Louis, MO, USA). The Muller–Hinton agar (MHA) was obtained from BD Difco^TM^ (Sparks, Los Angeles, CA, USA). All the cell cultures, fetal bovine serum, penicillin-streptomycin, and Dulbecco’s Modified Eagle Medium (DMEM) were of analytical grade and obtained from Thermo Fisher Scientific (Waltham, MA, USA).Fluorescent stains (ethidium bromide (EB) and acridine orange (AO), rhodamine 123 (Rh123), and dichlorofluorescein diacetate (DCFH-DA)) were obtained from Sigma-Aldrich (St. Louis, MO, USA).

### 2.1. Synthesis of S-AgNP NPs

Different concentrations of starch (0.5%, 1.0%, 1.5%, 2.0%) were dissolved in 0.4 M NaOH. After the starch was fully dissolved, 5 mM sodium nitroprusside and 5 mM silver nitrate were added (1:1 *v*/*v*) [15]. The reaction solutions were mixed well and homogenized under dark conditions for 24 h. The precipitate from the solution was collected as S-AgNP NPs after centrifugation (4000 rpm, 10 min). The best concentration of starch for the S-AgNP NPs was determined by the antibacterial activity. AgNP NPs were synthesized according to the previously reported method with slight modifications. Then, 5 mM sodium nitroprusside and 5 mM silver nitrate were mixed in the same ratio. The solution was completely mixed and centrifuged under dark conditions. The precipitate obtained was dried as AgNP NPs.

### 2.2. Characterization

The micromorphology and elemental content of S-AgNP NPs were analyzed by transmission electron microscopy (FE-TEM) and energy-dispersive X-ray spectroscopy (EDS) (JEOL-JSM, Akishima, Japan). The particle size distributions and potentials of the nanoparticles were determined by a zeta potential particle size analyzer (Malvern PANalytical, Worcestershire, UK). The crystallinity of the starch, S-AgNP NPs, and AgNP NPs were determined using X-ray powder diffraction (XRD, X’pert-pro MPD-PANalytical, Worcestershire, UK). The functional characteristics of the nanoparticles were detected by Fourier-transform infrared spectroscopy (FTIR, PerkinElmer Paragon 500, Waltham, MA, USA). For the XRD and FTIR analysis, the starch, S-AgNP NPs, and AgNP NPs were analyzed after sufficient drying and grinding into powder. The silver content of S-AgNP NPs was analyzed using ICP-MS (PerkinElmer (NextION 300D), Waltham, MA, USA); in brief, 10 mg of sample was dissolved in 150 μL of HNO_3_ and 350 μL of HCl, and after 3 h of full acid digestion, 20 μL was diluted in 10 mL of 0.2% HNO_3_ solution. The silver and iron contents of AgNP NPs were determined according to standard calibration curves for silver and iron (1, 2.5, 5, 10, 25, 50, 100 μg/L) [16].

### 2.3. Well Diffusion Assay

The inhibitory activity of S-AgNP NPs against a wide range of bacterial pathogens was determined by a well diffusion assay. Briefly, the bacterial pathogens were *B. cereus*, *L. monocytogenes*, *S. aureus*, *E. coli*, and *S. enterica*. Bacterial pathogens were first incubated in MHB medium at 37 °C for 24 h. The cultured bacterial liquid (50 μL) was evenly spread on the MHA medium. The medium was punched by a metal cork borer injected with the sample solution and incubated for 12 h. Each zone of inhibition was measured three times and photographed. Tetracycline hydrochloride (TCH) was used as a positive control.

### 2.4. MIC and MBC Value

To further evaluate the inhibitory activity of S-AgNP NPs against bacterial pathogens, the minimum inhibitory concentration (MIC) and minimum bactericidal concentration (MBC) of nanoparticles against five pathogens were determined [17]. Briefly, the bacterial solution after 24 h of incubation was injected in 96-well plates with different final concentrations of S-AgNP NPs (1.9 to 250 μg/mL). The growth curves of the bacteria were determined at 2, 4, 6, 8, 12, and 24 h, respectively.

### 2.5. Antibiofilm Assay

Bacterial biofilms can form in wounds and affect the rate of wound healing. To test the inhibitory activity of S-AgNP NPs on bacterial biofilms, five pathogens were tested. For testing, 100 μL overnight cultures of bacteria were supplemented with an equal volume of MHB medium mixed with S-AgNP NPs. The final concentration was between 1.9 to 250 μg/mL. After incubating the plates for 24 h at 37 °C, the planktonic bacteria were removed, and the residual bacteria were washed with PBS. Adherent biofilms in the plates were fixed with methanol and stained with 0.1% crystal violet for 30 min. After washing and drying the dye completely, it was dissolved in 95% ethanol, and the absorbance value was measured at 590 nm [18].

### 2.6. Antioxidant Assay

The antioxidant and reducing capacities of S-AgNP NPs were determined in a 1,1-diphenyl-2-picryl-hydrazyl (DPPH) and 2,2′-Azino-bis (3-ethylbenzothiazoline-6-sulfonic acid) (ABTS^+^) radical scavenging antioxidant assay [19]. Briefly, the DPPH assay was prepared with DPPH dissolved in methanol directly. For the ABTS assay, ABTS (7 mM) was added to the potassium persulfate solution (2.45 mM) and kept in the dark for 24 h. Additionally, the S-AgNP NPs were dissolved in methanol. Then, 100 μL of the sample was mixed with the same volume of ABTS^+^ and DPPH solutions and reacted for 30 min, and the OD values were measured at 734 nm and 517 nm, respectively. The free radical scavenging rate was calculated by comparing it with the control group.

### 2.7. Cytotoxicity

The cytotoxicity of nanoparticles on mouse embryonic fibroblast cell line NIH3T3 cells was determined by using the WST kit [20]. NIH3T3 cells were procured from a Korean cell line bank (Seoul National University College of Medicine, Seoul, Republic of Korea). Briefly, NIH3T3 cells were cultured for two days in a DMEM medium supplemented with penicillin-streptomycin solution (1%) and 10% FBS. After the cells reached 80% coverage in the culture plate, the cells were activated with DMEM medium and homogenously inoculated into 96-well plates with fresh medium overnight. Cells were treated with different concentrations of nanoparticles and incubated for 24 h, and final concentrations were obtained (1000, 500, 250, 125, 62. 5, 31.3, 15.6, 7.8, 3.9, 1.9 μg/mL). Then, 10 μL of WST was used to stain the cells for 1 h, and the absorbance value was read at 450 nm. Cell viability was calculated according to the following formula:$$\mathrm{Cell}\;\mathrm{viability}\;(\%)=\frac{{\mathrm{OD}}_{\mathrm{sample}}}{{\mathrm{OD}}_{\mathrm{control}}}\times 100$$

### 2.8. Cell Fluorescent Staining Assay

AO/EB staining assay was performed using a previously reported protocol [21]. Samples consisted of cells that were processed for 24 h, after which the cells were washed with PBS and removed. Then, 10 μL dye mixture (1:1) was subjected to AO/EB staining. The cell morphology was observed under a fluorescence microscope. The DCFH-DA was used to assess the ROS expression levels in control and S-AgNP NPs-treated NIH3T3 cells. After cells were grown to 80% in 24-well plates, NIH3T3 cells were washed with PBS after incubation to remove excess dead cells and medium. Then, cells were stained by using DCFH-DA for 15 min and observed under the fluorescence microscope [22]. Rh123 associated with changes in cellular mitochondrial membrane potential. Sample-treated cells were cultured for 24 h. Then, the floating cells were removed and washed with PBS. The cells were stained by adding 10 μL of Rh123 for 15 min, and Rh123 was washed off. Cells were then observed under a fluorescence microscope and photographed [23].

### 2.9. In Vitro Wound Healing Assay

The in vitro scratch assay of S-AgNP NPs and AgNP NPs was performed according to the previously reported protocol [24]. For this purpose, NIH3T3 (1 × 10^6^ cells/mL) were inoculated into 12-well plates with complete DMEM and incubated at 37 °C, 5% CO_2_. As soon as the cell growth reached a homogeneous monolayer, the cells were scratched with a sterilized pipette tip and then washed with PBS to remove excess detached cells. Subsequently, S-AgNP NPs (62.5 µg/mL in PBS) and AgNP NPs (7.8 µg/mL) were added to the NIH3T3 cells. In the control group, cells were untreated. After 0, 12, 24, 36, 48, and 72 h of treatment, images were taken by a microscope. In addition, the initial and final widths of the scratches were measured using ImageJ software (ImageJ bundled with 64-bit Java 1.8.0_172). Scratch shrinkage was calculated according to the following formula [25].
$$\%\;\mathrm{of}\;\mathrm{wound}\;\mathrm{healing}=\frac{\mathrm{wound}\;\mathrm{area}\;\mathrm{on}\;\mathrm{the}\;\mathrm{initial}\;\mathrm{day}\;-\mathrm{wound}\;\mathrm{area}\;\mathrm{on}\;\mathrm{the}\;\mathrm{final}\;\mathrm{day}}{\mathrm{wound}\;\mathrm{area}\;\mathrm{on}\;\mathrm{the}\;\mathrm{initial}\;\mathrm{day}}\times 100$$

### 2.10. Hemolysis Assay

The biocompatibility of S-AgNP NPs with blood was determined by hemolysis assay following the earlier protocol but with a few modifications [26]. In brief, red blood cells (RBCs) were prepared from defibrinated blood and dissolved in PBS. Different concentrations of samples (200 μL) and RBCs (200 μL) were incubated for 1 h and 24 h at 37 °C. Where triton X-100 was used as the positive control, and PBS was used as the negative control. After treatment, the RBCs were centrifuged and precipitated at the bottom of the centrifuge tube, and the hemolytic activity was determined by the absorbance of the supernatant at 545 nm.
Hemolysis (%)=(“OD sample” −”OD negative”)/(“OD positive” −”OD negative”)×100

### 2.11. Statistical Analyses

Each experiment was repeated three times, and the results were expressed as $\bar{x}$ ± standard deviation. To assess the statistical significance of the results, a one-way analysis of variance (ANOVA) was performed. The *p*-values less than 0.05 were considered statistically significant.

## 3. Results and Discussion

### 3.1. Preparation of S-AgNP NPs

In pursuit of enhanced inhibitory characteristics against pathogens, S-AgNP NPs were fabricated, employing varying concentrations of starch. The synthesis process involved the amalgamation of starch with silver nitrate and sodium nitroprusside for 2 h. This resulted in the 1.5% starch solution manifesting a dark hue with conspicuous black precipitates. Subsequently, the antibacterial efficacy of S-AgNP NPs, synthesized under diverse starch concentrations, was systematically assessed. The outcomes, as depicted in Appendix A, unequivocally demonstrated that S-AgNP NPs exhibited maximal inhibitory potency against all pathogens when the starch concentration was set at 1.5%. Consequently, the focal point of this investigation was the exploration of S-AgNP NPs synthesized within a 1.5% starch solution.

### 3.2. Characterization of S-AgNP NPs

#### 3.2.1. TEM Analysis and Total Ag Content of S-AgNP NPs

The detailed morphological features of S-AgNP NPs were observed by TEM assay. The results (Figure 1a) show that a large number of S-AgNP NPs are clustered in the same region. However, previous studies reported that AgNP NPs had distinct cubic structures and were able to observe prismatic corners in TEM images [14]. A comparison of AgNP NPs revealed that S-AgNP NPs also have a partial cubic structure. However, S-AgNP NPs became spherical due to the starch fabrication. It is hypothesized that starch influences the morphology of the AgNP NPs, resulting in a less cubic morphology. The starch after ultrasonic extension showed obvious swelling and a large agglomeration phenomenon, leading to a large number of nanoparticle aggregations [27], corresponding to the close aggregation phenomenon of S-AgNP NPs observed in the TEM analysis. The elements enriched in S-AgNP NPs were analyzed by EDS (Figure 1b–f). The S-AgNP NPs were found to contain a large amount of Ag elements, followed by Fe, C, O, and N elements (Figure 1h). This indicates that silver ions replace sodium ions in sodium nitroprusside complexes, resulting in the formation of S-AgNP NPs [14]. Analysis of the ICP-MS data showed the formation of detectable silver ions (0.20 ± 0.21 µg/mg) in the S-AgNP NPs (Table 1), which is also consistent with the EDS results.

#### 3.2.2. Zeta Size and Zeta Potential of S-AgNP NPs

The rheology and stability of nanoparticles are related to the hydrodynamic size and potential of the nanoparticles. The results (Table 1) show that the average particle size of S-AgNP NPs was 356 ± 22.28 d. nm, and the PDI value was 0.368 ± 0.02. The zeta potential was −27.8 ± 2.80 mV. However, AgNP NPs had a particle size of 171 ± 8.26 d. nm, a PDI value of 0.157 ± 0.01, and a potential of −13.82 ± 1.06 mV. In a previous study, it was found that AgNP NPs have a particle size of 260 ± 15 d. nm and a potential of −16.6 ± 3 mV [14]. It can be seen that AgNP NPs are negatively charged. Therefore, compared to AgNP NPs, the absolute value of the potential of S-AgNP NPs is greater, indicating the enhanced particle dispersion stability of S-AgNP NPs in the aqueous phase. The increased particle size compared to AgNP NPs may be attributed to the effects of the starch.

#### 3.2.3. XRD Analysis of S-AgNP NPs

The crystalline properties of S-AgNP NPs were assessed in XRD analysis. The results (Figure 2a) of S-AgNP NPs exhibited strong characteristic diffraction peaks at 2 theta = 38.31°, while the weaker characteristic diffraction peaks were at 14.22°, 17.33°, 19.20°, 25.50°, 28.32°, 30.57°, 32.67°, 66.84°, and 77.95°. The starch has characteristic peaks at 15.02°, 17.25°, 18.16°, and 23.10°, which indicates the semi-crystalline structure of starch [28]. Among them, the derived peak of S-AgNP NPs at 17.20° is most likely owing to starch characteristics. AgNP NPs have special derivatization peaks at 13.80°, 19.18°, 21.59°, 24.6°, 25.39°, 30.15°, 33.35°, 36.34°, and 38.97°, while 13.80°, 19.18°, 21.59°, and 25.39° correspond to the derivatized surfaces of (001), (010), (110), and (210), respectively [29]. The characteristic peaks of these AgNP NPs also appeared in the XRD diffraction patterns of S-AgNP NPs corresponding to 14.22°, 19.20°, and 25.50°, respectively. Therefore, it can be inferred that S-AgNP NPs possess the crystal structure from starch and AgNP NPs. Notably, the strong peak from AgNP NPs located at 19.18°, corresponding to sodium nitroprusside, was not represented in S-AgNP NPs. The concentration of starch affects the display of the characteristic peaks of the nanoparticles. The significance of the characteristic peak of iron tetroxide decreased when starch was at 0.5% concentration [30]. Thus, it is assumed that the starch and iron in sodium nitroprusside produced a combination that caused the distinctive peak to disappear [31]. However, at 38.31°, the S-AgNP NPs showed characteristic peaks belonging to the FCC structure of silver, suggesting that starch and sodium nitroprusside can promote the reduction of large amounts of silver ions [32].

#### 3.2.4. FTIR Analysis of S-AgNP NPs

The surface functional group structures of S-AgNP NPs were examined by FTIR. The results (Figure 2b) showed that S-AgNP NPs had distinct peaks at 579.84, 991.23, 1424.23, 2014.81, and 3324.62 cm^−1^, where 3324.62 cm^−1^ corresponds to the OH stretching vibration of the starch structure and the peak at 3295.28 cm^−1^ in starch [33]. In addition, the peak located at 2014.81 cm^−1^ from the S-AgNP NPs corresponds to the vibrational frequency of the equatorial -CN and correlates to 2177.77 cm^−1^ in the AgNP NPs [29].

The peak at 1648.67 cm^−1^ corresponds to the H-bond linked to the carbohydrate in starch. The other peak at 991.23 cm^−1^ corresponds to the C-O stretching motion and also corresponds to 992.59 cm^−1^ in starch. The FTIR spectra of AgNP NPs showed major peaks at 3851.04, 2177.77, 1938.04, 662.43, 514.28, and 427.53 cm^−1^, where 3851.04 cm^−1^ corresponds to an overtone of the -NO stretching vibration [15]. There are some defined peaks below 1000 cm^−1^, which are 427.53, 514.28, and 662.43 cm^−1^, corresponding to ferricyanide [34,35]. The peak of ferrocyanide also appeared in the S-AgNP NPs, corresponding to 579.84 cm^−1^. Overall, specific peaks from starch and AgNP NPs can be detected from S-AgNP NPs (for instance, 579.84, 991.23, 2014.81, and 3324.62 cm^−1^). These results reveal that S-AgNP NPs have structures partially derived from starch and AgNP NPs.

### 3.3. Antibacterial Activity of S-AgNP NPs

Wound healing is often accompanied by bacterial infection, which induces a variety of inflammatory conditions and leads to the formation of chronic wounds. The antibacterial activity of S-AgNP NPs was determined by Gram-positive and Gram-negative bacteria. Overall, the results (Figure 3) showed that S-AgNP NPs displayed a broad spectrum of antibacterial activity against both Gram-positive and Gram-negative bacteria. The S-AgNP NPs exhibited clear inhibition zones (Table 2), measured at 6.67 ± 0.20 mm, 6.33 ± 0.24 mm, 6.50 ± 0.40 mm, 6.10 ± 0.20 mm, and 6.67 ± 0.47 mm for *B. cereus*, *L. monocytogenes*, *S. aureus*, *E. coli*, and *S. enterica*, respectively. In addition, the corresponding inhibition zones of AgNP NPs were measured at 7.50 ± 0.40 mm, 10.67 ± 0.47 mm, 9.67 ± 0.20 mm, 9.80 ± 0.20 mm, and 10.23 ± 0.40 mm. Nevertheless, S-AgNP NPs showed less antibacterial activity as compared to AgNP NPs.

To further test the antibacterial activity of S-AgNP NPs, the MIC and MBC were tested. It was found that S-AgNP NPs had the strongest bactericidal ability for *L. monocytogenes* and started to inhibit the growth of *L. monocytogenes* at the concentration of 15.6 μg/mL (Figure 4b). The S-AgNP NPs showed the weakest inhibitory ability against *S. enterica*. The MIC value was 62.5 μg/mL, and the MBC value was 125 μg/mL. Regarding *S. aureus*, S-AgNP NPs started to inhibit at 31.3 μg/mL and completely inhibited pathogen growth at 62.5 μg/mL (Figure 4b). Similarly, S-AgNP NPs were able to inhibit and kill *E. coli* at 31.3 μg/mL and 62.5 μg/mL, respectively. For *B. cereus* (Table 3), S-AgNP NPs inhibited bacterial growth at 15.6 μg/mL and completely inhibited the pathogen at 125 μg/mL. AgNP NPs have significant bacteriostatic activity against both Gram-positive and negative bacteria at very low concentrations. It was less than 2 μg/mL against both *E. coli* and *B. cereus* [14]. Starch itself has no antibacterial activity. However, as a capping agent, starch has been repeatedly proven to stabilize the released silver ions [36]. The antibacterial properties of S-AgNP NPs are speculated to be similar to AgNP NPs and come from the release of silver ions from the nanoparticles. The released silver ions lead to antimicrobial activity mainly from oxidative radical generation pressure and consequent DNA damage [14]. Inactivation of mitochondrial enzymes within bacteria, denaturation of structural proteins and cytoplasm, and structural changes in cell membranes have all been linked to silver ions. Unlike Gram-positive bacteria, silver ions also intervene in signaling in Gram-negative bacteria [37]. AgNP NPs have been shown to have the ability to cause morphological changes in bacteria, disrupting the bacterial membrane and leading to cytoplasmic efflux and ultimately bacterial death [29]. Hence, S-AgNP NPs may have a similar function as prospective bacteriostatic agents during the wound-healing process.

### 3.4. Anti-Biofilm Activity of S-AgNP NPs

When a wound is infected, pathogens can form biofilms in the wound, affecting the rate of healing and the healing process. Therefore, the inhibitory activity of S-AgNP NPs against biofilms generated by different pathogens was determined. The results show (Figure 5a) that S-AgNP NPs were able to inhibit biofilms produced by five pathogens. Among them, S-AgNP NPs had the strongest biofilm inhibition activity for *B. cereus* (Figure 5b). The biofilm inhibition rate of *B. cereus* reached 91.7% at the highest concentration (250 μg/mL). S-AgNP NPs had lower biofilm inhibition activity against *E. coli* at the same concentration (47.4%). For *L. monocytogenes*, S-AgNP NPs were able to inhibit 83.03% at 250 μg/mL, and the strong inhibition continued up to 7.8 μg/mL (77.31%). At 3.9 μg/mL, biofilm inhibition was reduced to 57.60%. For *S. enterica*, S-AgNP NPs were able to inhibit ~80% of the biofilm at a lower concentration (15.6 μg/mL). For *S. aureus*, S-AgNP NPs inhibited ~80% of the biofilm at high concentrations but completely failed to inhibit at low concentrations. Biofilms contribute to the deterioration of the wound environment, negatively affecting wound healing by inhibiting cell proliferation and migration and upregulating pro-inflammatory cytokines [38]. Currently, no study has found that AgNP NPs can inhibit bacterial biofilm formation. It is presumed that the biofilm inhibitory capacity of S-AgNP NPs is also related to the release of silver ions. Silver ions interact with biofilm nucleic acids, and the nanoparticles themselves interact with a wide range of proteins and polysaccharides in biofilms [39]. These results all reveal that S-AgNP NPs have inhibitory biofilm activity and possess potential as biofilm scavengers.

### 3.5. Antioxidant Properties of S-AgNP NPs

The antioxidant capacity of S-AgNP NPs was determined by both ABTS^+^ and DPPH radicals. The results (Figure 6) showed that the free-radical-scavenging capacity of S-AgNP NPs was positively correlated with the concentration. At 125 μg/mL, S-AgNP NPs were able to scavenge the vast majority of DPPH radicals. For DPPH radicals and ABTS^+^ radicals, the scavenging activities of S-AgNP NPs were significantly different, and the IC_50_ concentrations were 66.8 μg/mL and 213.8 μg/mL, respectively. Only at 500 μg/mL were S-AgNP NPs able to scavenge the most ABTS^+^ radicals. At other concentrations, the scavenging activity was significantly reduced. There are no studies that have found antioxidant activity in silver nitroprusside nanoparticles. However, one study reported that starch has been found to have 74% scavenging activity of DPPH radicals at 100 μg/mL [40]. It has also been reported that starch was found to have low antioxidant activity, with a scavenging rate of about 35.59% for DPPH free radicals [41]. In general, starch synthesized from phenolic compounds and flavonoid-rich plant extracts have shown high scavenging activity. Additionally, the antioxidant activity of S-AgNP NPs may also be related to silver ions. Since silver ions have two oxidation states, Ag^+^ and Ag^2+^ [42], they can act as antioxidants through single electron and hydrogen atom transfer [43].

### 3.6. Cell Viability and Cell Staining

The cytotoxicity of S-AgNP NPs was measured with normal NIH3T3 cells. The results (Figure 7a) of toxicity assays revealed that both S-AgNP NPs and AgNP NPs showed significant cytotoxicity for NIH3T3 cells. Interestingly, S-AgNP NPs were significantly reduced in cytotoxicity compared to AgNP NPs. At higher concentrations (>500 μg/mL), both S-AgNP NPs and AgNP NPs showed strong cytotoxicity. The cell viability at this time was below 20% in both of them. Along with the decrease in concentration, the cytotoxicity of S-AgNP NPs began to decrease. On the contrary, AgNP NPs still showed extremely strong cytotoxicity. At 62.5 μg/mL, S-AgNP NPs showed no cytotoxicity, and the cell viability reached 93.46% at this time, while the cell viability of AgNP NPs was 9.74%. Against NIH3T3 cells, the cytotoxicity of AgNP NPs remained above 1.9 μg/mL. Starch stabilized the nanoparticles and significantly reduced the cytotoxicity.

The results of cell fluorescence staining of S-AgNP NPs were similar to the cytotoxicity results (Figure 7b). In AO/EB staining, the green color represents complete, non-apoptotic cells with high cell viability. Orange or red fluorescence represents the loss of cell viability, cell fragmentation, and apoptosis. Against NIH3T3 cells, S-AgNP NPs (62.5 μg/mL) did not cause apoptosis and cell disruption. In comparison to the control group, NIH3T3 cell density was not significantly reduced, and cell viability was maintained at a strong level. AgNP NPs (62.5 μg/mL) were able to cause a reduction in cell density, and a large number of cells were fragmented or died. DCFH-DA correlates with intracellular ROS levels. Strong green fluorescence represents an increase in the level of intracellular ROS, and the cell was in a state of senescence and apoptosis. On the other hand, weak green fluorescence represents low levels of intracellular ROS and high cell viability. In contrast to the control group, the S-AgNP NPs-treated cells showed weaker green fluorescence. On the contrary, the morphology of NIH3T3 cells treated by AgNP NPs was changed, the number of cells was decreased, and the intracellular ROS content was increased. To detect the effect of S-AgNP NPs on the mitochondrial membrane potential of NIH3T3, the cells were stained for fluorescence using Rh123. It was found that S-AgNP NPs-treated cells showed a strong red light. This represents the mitochondrial membrane integrity of the NIH3T3 cells. On the contrary, AgNP NPs-treated cells showed reduced fluorescence intensity and decreased cell number. This indicates that the mitochondrial membrane of NIH3T3 cells has reduced transport capacity and electronegativity, so the ability of cellular mitochondria to accumulate Rh123 was also lost. Nanoparticles may cause cytotoxicity due to their size, shape, chemical or physical properties, and high reactivity [44]. Starch nanoparticles from corn have antioxidant and anti-inflammatory properties and are less toxic to human cell lines [45]. Therefore, it was inferred that starch may stabilize the nanoparticles and protect the cells, reducing the cytotoxicity of the nanoparticles. Hence, S-AgNP NPs can be applied as a potential antibacterial wound-healing agent with no toxicity.

### 3.7. Wound Healing

In vitro wound healing assay, cell migration, and proliferation in mechanical wounds with and without sample treatment were assessed. A comparison of different time intervals (0, 12, 24, 36, 48, and 72 h) with sample treatments (S-AgNP NPs, AgNP NPs) as well as control is shown in Figure 8. Affected by S-AgNP NPs, NIH3T3 cells proliferated and migrated to the damaged area more efficiently compared to the control. The ability of NIH3T3 cells to close the damage was enhanced at 12, 24, and 48 h. Finally, NIH3T3 cells were completely healed at 72 h. In addition, AgNP NPs also showed the ability to promote wound healing. Comparatively, the ability of S-AgNP NPs to promote cell migration and proliferation was significantly enhanced. As previously reported, AgNP NPs can accelerate wound healing in C57BL6 mice by altering the macrophages [14]. AgNP NPs combined with nanocrown ether-based porous organic polymers accelerated wound healing while stabilizing with bacteriostatic properties [29]. In addition, starch is often used as a nanofibrous scaffold to promote cell migration and accelerate cell migration and wound healing [46]. As a reducing agent and carrier of nanoparticles, starch can stabilize nanoparticles, reduce cytotoxicity, and exert the wound-healing ability of nanoparticles [47].

### 3.8. Hemolysis

The safety and hemocompatibility of S-AgNP NPs were determined by hemolysis assay. The results (Figure 9) showed that S-AgNP NPs were hemolytic only at 500 μg/mL, with a hemolysis rate of 13.20%. S-AgNP NPs had a hemolysis rate of less than 5% at all other concentrations (1.9 to 250 μg/mL). AgNP NPs caused hemolysis at both 250 and 500 μg/mL. The hemolysis rates were 5.45% and 8.64%, respectively. Starch does not cause significant hemolysis of blood cells. AgNP NPs have good biocompatibility at low concentrations [13]. In particular, the hemolysis rate of S-AgNP NPs at 250 μg/mL is significantly lower than that of AgNP NPs. After 24 h incubation, S-AgNP NPs were still able to show a lower rate of hemolysis than AgNP NPs (Appendix A). In general, the rate of hemolysis caused by nanoparticles is related to the particle size and concentration of the nanoparticles. The smaller the particle size, the stronger the hemolysis caused by the nanoparticles [48]. The negative charge on the nanoparticles may prevent the erythrocytes from interacting with the nanoparticles, which reduces hemolysis [49]. Starch causes blood cells to aggregate and acts as a protective agent for blood cells [50]. Overall, the weak hemolysis caused by S-AgNP NPs is attributed to their particle size and negative charge potential.

## 4. Conclusions

Currently, there is a high demand for agents that promote wound healing and inhibit bacteria in the treatment of chronic wounds. AgNP NPs have shown excellent properties in both antibacterial activities and wound healing. In this study, starch-NP NPs were synthesized by combining with starch. The S-AgNP NPs were characterized by FE-TEM, FTIR, and XRD, and it was found that the S-AgNP NPs possessed functional groups and crystal structures from starch and AgNP NPs. The particle size was 356 ± 22.28 d.nm, and the potential was −27.8 ± 2.80 mV. In addition, S-AgNP NPs had significant antibacterial and antibiofilm activities against five pathogens below 62.5 μg/mL. For DPPH and ABTS free radicals, S-AgNP NPs can scavenge most of the free radicals, and the IC_50_ concentrations were 66.8 μg/mL and 213.8 μg/mL, respectively. Moreover, S-AgNP NPs have good biocompatibility and low cytotoxicity. In addition, S-AgNP NPs can accelerate cell migration and proliferation while exerting their antibacterial activity at 62.5 μg/mL, which can promote wound healing. Overall, S-AgNP NPs possessed the physical and physiological activities of starch and AgNP NPs and had the potential to be used as antimicrobial wound-healing agents.

## Figures and Tables

**Figure 1 antioxidants-13-00154-f001:**
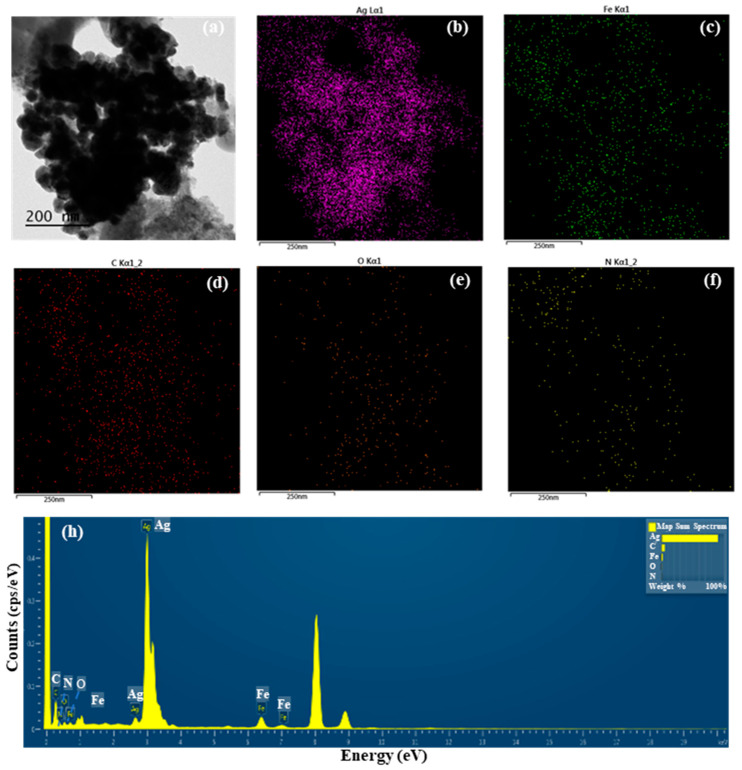
Transmission electron microscopic (TEM) image of starch–silver nitroprusside nanoparticles (S-AgNP NPs) (**a**); elemental (Ag, Fe, C, O, N) mapping of S-AgNP NPs (**b**–**f**); elemental spectrum of S-AgNP NPs (**h**).

**Figure 2 antioxidants-13-00154-f002:**
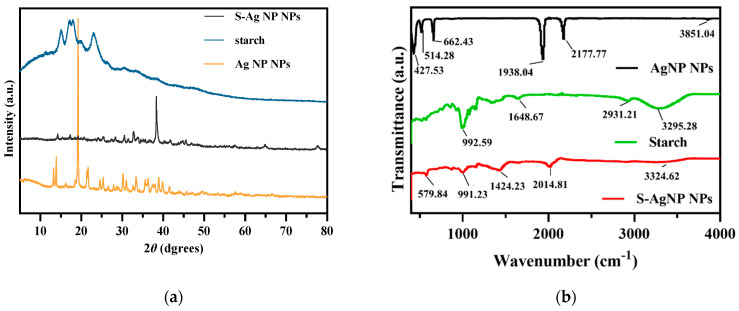
XRD (**a**) and FTIR (**b**) spectrum of S-AgNP NPs compared with AgNP NPs and starch.

**Figure 3 antioxidants-13-00154-f003:**
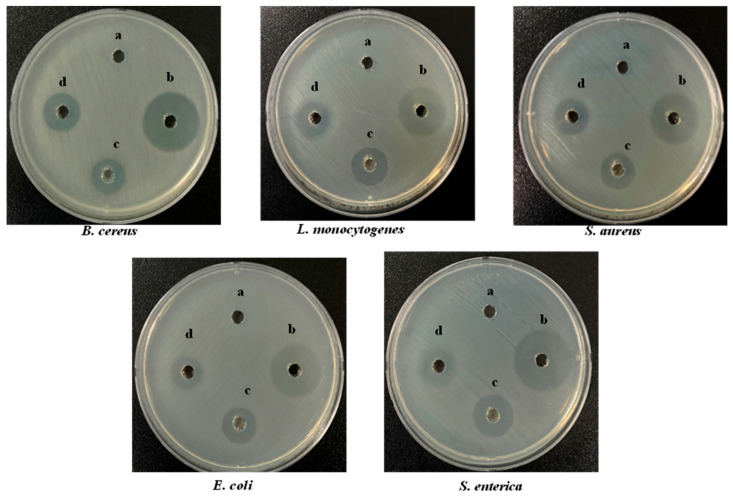
Antibacterial activity of S-AgNP NPs, AgNP NPs, and starch on different bacterial pathogens compared with tetracycline hydrochloride (TCH) (a-50 μg/50 μL starch; b-50 μg/50 μL TCH; c-50 μg/50 μL AgNP NPs; d-50 μg/50 μL S-AgNP NPs).

**Figure 4 antioxidants-13-00154-f004:**
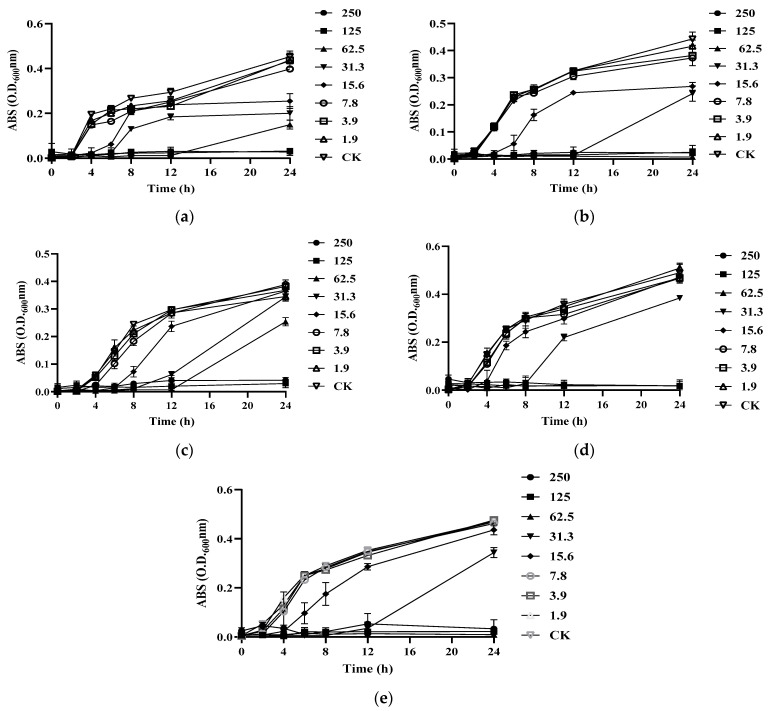
Growth curves of different bacteria treated by S-AgNP NPs: (**a**) *B. cereus*; (**b**) *L. monocytogenes*; (**c**) *S. aureus*; (**d**) *E. coli*; (**e**) *S. enterica*.

**Figure 5 antioxidants-13-00154-f005:**
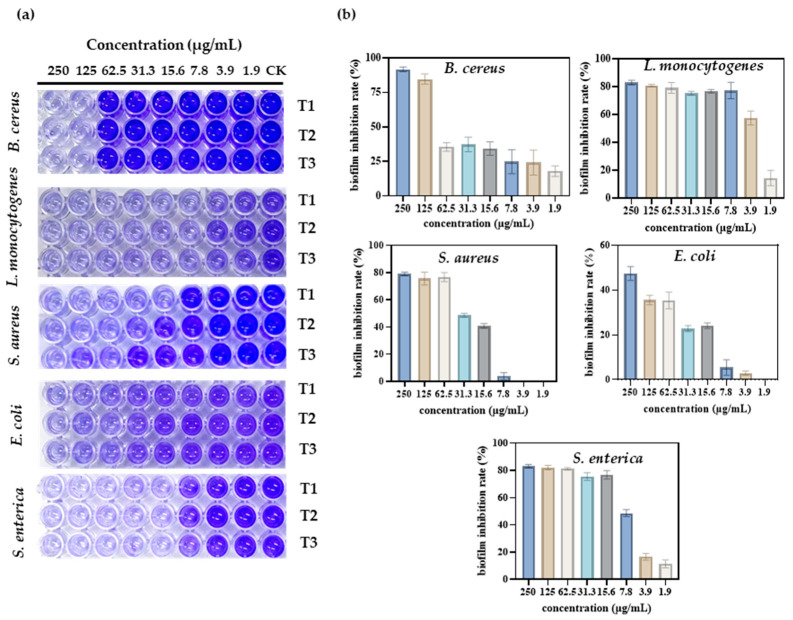
Antibacterial activity of different concentrations of S-AgNP NPs on different bacterial pathogens. Visual image of the anti-biofilm assay in a 96-well plate (**a**); biofilm inhibition rate of S-AgNP NPs (**b**). T1, T2, and T3 indicate each of the experiments’ triplicate results.

**Figure 6 antioxidants-13-00154-f006:**
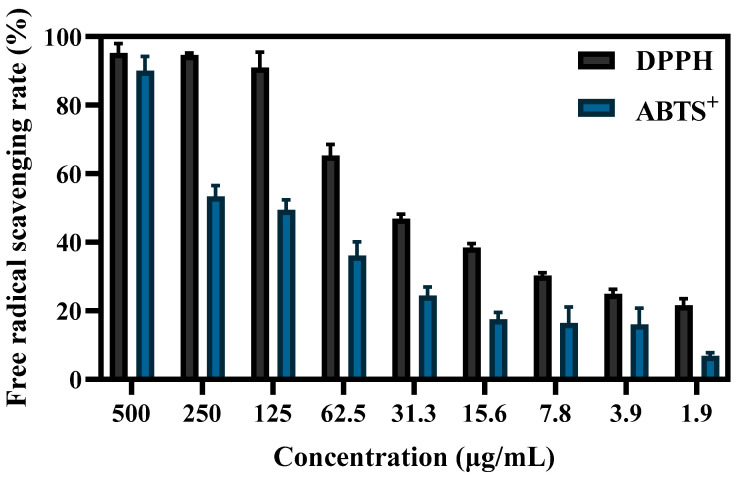
The DPPH and ABTS^+^ free-radical-scavenging ability of different concentrations of S-AgNP NPs.

**Figure 7 antioxidants-13-00154-f007:**
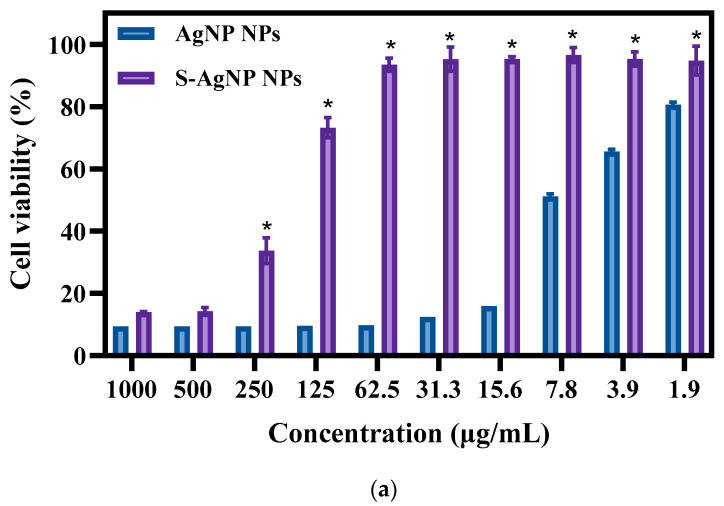
Cell viability (**a**) and fluorescent staining analysis (**b**) of S-AgNP NPs and AgNP-NP-treated NIH3T3 cells. * *p* < 0.05 represents a significant difference versus the AgNP NPs each time.

**Figure 8 antioxidants-13-00154-f008:**
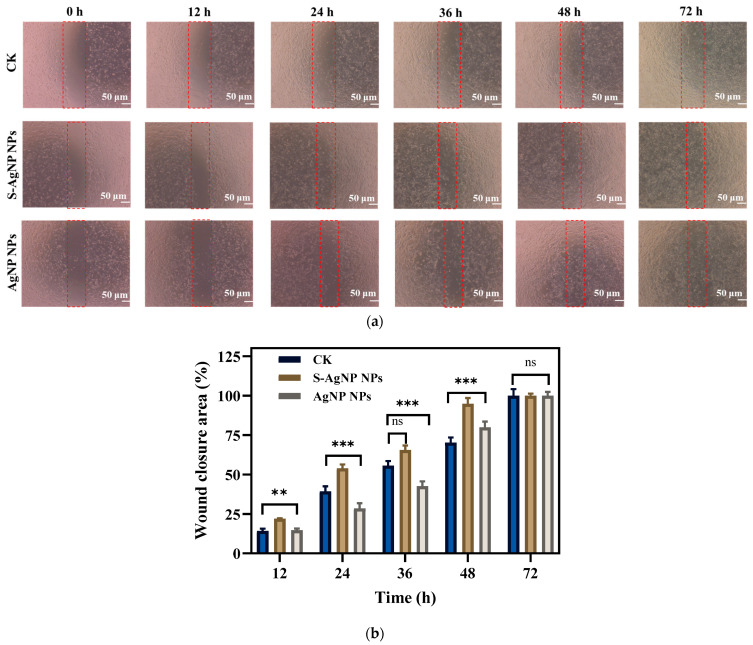
In vitro scratch wound healing assay of NIH3T3 cells in the presence of S-AgNP NPs and AgNP NPs monitored with light-inverted microscopy (**a**). The red dotted line represents the area of the wound scratch. Bar graph illustrating percentage wound closure at indicated time points (0, 12, 24, 36, 48, and 72 h after the initiation of the scratch when the cells were treated with the 62.5 μg/mL concentration of S-AgNP NPs and 7.8 μg/mL concentration of AgNP NPs during the scratch wound assay) (**b**). ** *p* < 0.01, *** *p* < 0.001 represents a significant difference between control, S-AgNP NPs, and AgNP NPs each time. ns means no significance.

**Figure 9 antioxidants-13-00154-f009:**
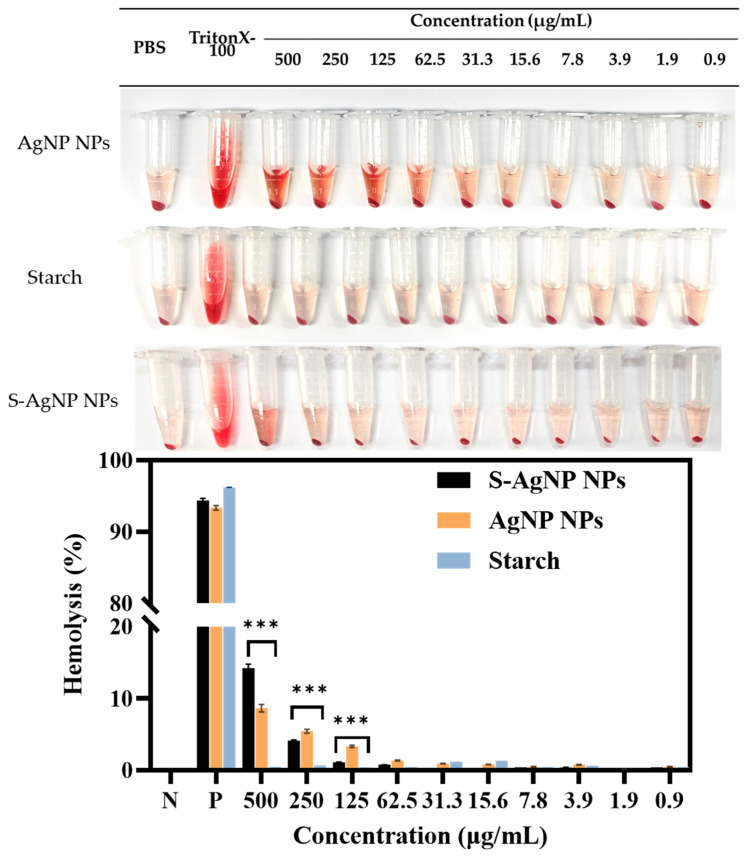
Hemolysis properties of S-AgNP NPs and AgNP NPs. *** *p* < 0.001 represents a significant difference between starch, S-AgNP NPs, and AgNP NPs each time.

**Table 1 antioxidants-13-00154-t001:** Hydrodynamic particle size, polydispersity index (PDI), zeta potential analysis and total Ag Content of S-AgNP NPs and AgNP NPs.

	Particle Size (d. nm)	Potential (mV)	PDI	Ag element Content (μg/mg)
S-AgNP NPs	356 ± 22.28	−27.8 ± 2.80	0.368 ± 0.02	0.20 ± 0.21
AgNP NPs	171 ± 8.26	−13.82 ± 1.06	0.157 ± 0.01	-

**Table 2 antioxidants-13-00154-t002:** Antibacterial activity of starch, S-AgNP NPs, AgNP NPs, and TCH.

Zone of Inhibition (mm)				
	Starch	S-AgNP NPs	AgNP NPs	TCH
*B. cereus*	0.00 ^d^	6.67 ± 0.20 ^c^	7.50 ± 0.40 ^b^	19.10 ± 0.20 ^a^
*L. monocytogenes*	0.00 ^d^	6.33 ± 0.24 ^c^	10.67 ± 0.47 ^b^	18.80 ± 0.30 ^a^
*S. aureus*	0.00 ^d^	6.50 ± 0.40 ^c^	9.67 ± 0.20 ^b^	18.30 ± 0.40 ^a^
*E. coli*	0.00 ^d^	6.10 ± 0.20 ^c^	9.80 ± 0.20 ^b^	19.20 ± 0.20 ^a^
*S. enterica*	0.00 ^d^	6.67 ± 0.47 ^c^	10.23 ± 0.40 ^b^	19.1 ± 0.20 ^a^

Values followed by the superscript letters in the column significantly differ (*p* < 0.05) among starch, S-AgNP NPs, AgNP NPs, and TCH.

**Table 3 antioxidants-13-00154-t003:** The MIC and MBC of S-AgNP NPs against different bacterial pathogens.

	MIC (μg/mL)	MBC (μg/mL)
*B. cereus*	15.6	125
*L. monocytogenes*	15.6	62.5
*S. aureus*	31.3	62.5
*E. coli*	31.3	62.5
*S. enterica*	62.5	125

## Data Availability

Data is contained within the article or Appendix A.

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
