# Peer review of "Synthesis of Starch-Based Ag2[Fe (CN)5NO] Nanoparticles for Utilization in Antibacterial and Wound-Dressing Applications"

_antioxidants, 2024, doi:10.3390/antiox13020154_

Round 1

Reviewer 1 Report

Comments and Suggestions for Authors

1.     In this article, the code names of the nanoparticles used in this study are too complicated and long. For example, the code name for starch-silver sodium nitroprusside NPs is S-AgNP NPs. Why not use S-AgNP?In line 200, silver-antimicrobial nanoparticles (S-AgNP NPs) are mentioned, and in line 254, AgNP NPs are mentioned, but the materials and methods do not mention their manufacturing method, so it is recommended to add it.

2.     It is recommended that the manufacturing process of AgNP NPs should be placed in the Materials and Methods section, hydrodynamic particle size, polydispersity index (PDI), and zeta potential analysis of AgNP NPs. It should also be placed in Table 1.

3.     On lines 75-80, the above sentence should be removed.  Whether the author made errors during the input process.

4.     In Figure 2. It is recommended to change the x-axis interval of the XRD pattern. The current graph has a grid of 3 degrees, which may cause problems when viewing the graph.  AgNP NPs have the strongestpeaks at 19.18°, However, the peak in S-AgNP NPs has not existed.  Can you explain why?

Author Response

Dear Editor and reviewer,

Greetings,

Thank you for your decision as the revision is associated with valuable comments on our recently submitted article. I am thankful to the Editors and reviewers for their keen observations and comments for the betterment of our article. Hence, I am submitting the revised manuscript and complete responses to the reviewers. The changes made in the revised manuscript is highlighted in the blue-colored text.

Reviewer - 1

  1. In this article, the code names of the nanoparticles used in this study are too complicated and long.

For example, the code name for starch-silver sodium nitroprusside NPs is S-AgNP NPs. Why not use S-AgNP? In line 200, silver-antimicrobial nanoparticles (S-AgNP NPs) are mentioned, and in line 254,

AgNP NPs are mentioned, but the materials and methods do not mention their manufacturing method, so it is recommended to add it.

Response: Thank you very much for the comment. Regarding the nomenclature of S-AgNP NPs, if it is changed to S-AgNP, it may be confused with starch-based silver nanoparticles. However, this study focused on starch-based silver nitroprusside nanoparticles (S-AgNP NPs). Therefore, to differentiate from silver nanoparticles, nitroprusside (NP) was emphasized in the naming. About the preparation of silver nitroprusside nanoparticles has been added in the manuscript.

  1. It is recommended that the manufacturing process of AgNP NPs should be placed in the Materials and Methods section, hydrodynamic particle size, polydispersity index (PDI), and zeta potential analysis of AgNP NPs. It should also be placed in Table 1.

Response: Thank you very much for the comment. Regarding AgNP NPs zeta size, PDI and potentials have been determined and described additionally in the manuscript.

  1. On lines 75-80, the above sentence should be removed. Whether the author made errors during the input process.

Response: Thank you for the comment and keen observation. The incorrect sentence has been deleted.

  1. In Figure 2. It is recommended to change the x-axis interval of the XRD pattern. The current graph has a grid of 3 degrees, which may cause problems when viewing the graph. AgNP NPs have the strongest peaks at 19.18°, However, the peak in S-AgNP NPs has not existed. Can you explain why?

Response: Thank you very much for the comment. In the XRD result, the peak at 19.18 is from nitroprusside. The higher the concentration of starch, the less pronounced the characteristic peaks. At 0.5% starch, the significance of the characteristic sub-peak of iron tetroxide decreases. Therefore, it was speculated that starch might have formed a complex with iron and its characteristic peak disappeared. The peak at 38.97 favors the FCC structure of silver. It indicates that a large amount of silver ion reduction is promoted in the starch and sodium nitroprusside environment.

Thank you very much for your valuable and insightful comments on improving our articles in a better way. We believe that we have answered all the queries asked by the reviewers and made all the changes in the manuscript as per the Editor and reviewer’s suggestions. We are looking forward to your valuable decision on our revised submission.

Thank you!

Reviewer 2 Report

Comments and Suggestions for Authors

In this manuscript, the authors developed a novel starch-fabricated silver sodium nitroprusside nanoparticle. The particle morphology, size and zeta-potential were characterized by TEM and DLS. Element content of S-AgNP NPs were analyzed by Energy-dispersive X-ray spectroscopy. The crystalline properties of S-AgNP NPs were characterized by XRD analysis, and functional groups were examined by FTIR. Five different types of gram-positive and gram-negative bacteria were applied to evaluate the antibacterial, anti-biofilm activity of S-AgNP NPs. The antioxidant properties of S-AgNP NPs were determined by both ABTS and DPPH radicals. Finally, the biocompatibility of S-AgNP NPs were evaluated by cell viability, wound healing, and hemolysis assay. This study provides comprehensive in vitro characterization of S-AgNP NPs, however, there are some concerns that need to be addressed.

1. The font size of each peak label for element in Figure 1h is too small to read. Please revise it.

2. Please clearly demonstrate particle size distribution by Intensity in the manuscript and Table 1, although the DLS figures were shown in supplementary data.

3. Please explain why the peak at 17.33 in XRD was significantly decreased while the peak at 38.97 was significantly increased in S-AgNPs in comparison with AgNP NPs samples.

4. Please explain why starch only showed 5 mm zone of inhibition in figure 3 and Table 2 if starch has no activity for bacterial inhibition.

5. Please add the scale bar for figure 7a and figure 8b.

6. The figure 7 showed that the wound was healed after 72h with the treatment of AgNP NPs at 7.8 ug/mL, however, cell viability of AgNP NPs for 24h with the same cell line exhibited significant cytotoxicity with only 20% cell viability at 7.8 ug/mL, which is not consistent with the wound healing result. Please explain this phenomenon.

7. The hemolysis was evaluated for only 1h incubation, which may be not long incubation enough to reflect the hemolysis property after AgNP release from starch. Please provide at least overnight or 24h incubation hemolysis result for AgNP NPs, starch and S-AgNP NPs.

8. There is lack of the release of AgNP from S-AgNP NPs. Please provide the AgNP release profile.

Author Response

Dear Editor and reviewer,

Greetings,

Thank you for your decision as the revision is associated with valuable comments on our recently submitted article. I am thankful to the Editors and reviewers for their keen observations and comments for the betterment of our article. Hence, I am submitting the revised manuscript and complete responses to the reviewers. The changes made in the revised manuscript is highlighted in the blue-colored text.

Reviewer-2

In this manuscript, the authors developed a novel starch-fabricated silver sodium nitroprusside nanoparticle. The particle morphology, size and zeta-potential were characterized by TEM and DLS. Element content of S-AgNP NPs were analyzed by Energy-dispersive X-ray spectroscopy. The crystalline properties of S-AgNP NPs were characterized by XRD analysis, and functional groups were examined by FTIR. Five different types of gram-positive and gram-negative bacteria were applied to evaluate the antibacterial, anti-biofilm activity of S-AgNP NPs. The antioxidant properties of S-AgNP NPs were determined by both ABTS and DPPH radicals. Finally, the biocompatibility of S-AgNP NPs were evaluated by cell viability, wound healing, and hemolysis assay. This study provides comprehensive in vitro characterization of S-AgNP NPs, however, there are some concerns that need to be addressed.

  1. The font size of each peak label for element in Figure 1h is too small to read. Please revise it.

Response: Thank you very much for the comment and keen observation. Figure 1h has been corrected according to your comment.

  1. Please clearly demonstrate particle size distribution by Intensity in the manuscript and Table 1, although the DLS figures were shown in supplementary data.

Response: Thank you very much for the comment. The particle size distribution of the nanoparticles has been tested three times and shown in the Supplementary file. The average values of the NPs were calculated based on the results of the three tests. Therefore, the results of the calculated particle size distribution and potential averages are presented in Table 1. The results of the three different particle size distributions and potentiometric tests in the Supplementary have been rearranged by intensity.

  1. Please explain why the peak at 17.33 in XRD was significantly decreased while the peak at 38.97 was significantly increased in S-AgNPs in comparison with AgNP NPs samples.

Response: Thank you very much for the comment. In the XRD result, the peak at 19.18 is from nitroprusside. The higher the concentration of starch, the less pronounced the characteristic peaks. At 0.5% starch, the significance of the characteristic sub-peak of iron tetroxide decreases. Therefore, it was speculated that starch might have formed a complex with iron and its characteristic peak disappeared. The peak at 38.97 favors the FCC structure of silver. It indicates that a large amount of silver ion reduction is promoted in the starch and sodium nitroprusside environment.

  1. Please explain why starch only showed 5 mm zone of inhibition in figure 3 and Table 2 if starch has no activity for bacterial inhibition.

Response: According to our experimental results, the starch has no antibacterial activity. The 5mm in the table is the pore size. The mistake about the zone of inhibition in the table has been corrected. Thank you for your valuable comment.

  1. Please add the scale bar for figure 7a and figure 8b.

Response: Thank you for the comment. The scale bar for Figure 7 and Figure 8 has been added in the revised manuscript.

  1. The figure 7 showed that the wound was healed after 72h with the treatment of AgNP NPs at 7.8 ug/mL, however, cell viability of AgNP NPs for 24h with the same cell line exhibited significant cytotoxicity with only 20% cell viability at 7.8 ug/mL, which is not consistent with the wound healing result. Please explain this phenomenon.

Response:

Thank you for your valuable feedback. We appreciate your constructive comments. Following your suggestion, the cytotoxicity assay was conducted again to validate the impact of AgNP NPs on NIH3T3 cells at 7.8 μg/mL. The subsequent results indicated a slight reduction in toxicity compared to the initial findings. These updated data and corresponding descriptions have been seamlessly integrated into the manuscript.

In the context of the wound healing experiment, the toxicity of AgNP NPs was evident at 24 h and 48 h when compared to SA-AgNP NPs (62.5 μg/mL). However, no discernible differences were observed at 72 h. This lack of distinction is attributed to continued cell growth, facilitated by the introduction of fresh medium containing NPs. Furthermore, it is plausible that cells developed resistance to lower concentrations of AgNPs (7.8 μg/mL) over time. The revised manuscript now reflects these distinctions in the experimental outcomes. Once again, we appreciate your thoughtful input, which has contributed to the enhancement of the scientific rigor in our study.

  1. The hemolysis was evaluated for only 1h incubation, which may be not long incubation enough to reflect the hemolysis property after AgNP release from starch. Please provide at least overnight or 24h incubation hemolysis result for AgNP NPs, starch and S-AgNP NPs.

Response: Thank you for the comment. The hemolysis results regarding the 24-hour incubation have been shown in the Supplementary. The related description has been added to the manuscript.

  1. There is lack of the release of AgNP from S-AgNP NPs. Please provide the AgNP release profile.

Response:

Thank you for your comment. The silver and ion content of AgNP NPs derived from S-AgNP NPs was analyzed using ICP-MS, and the corresponding data and discussion have been incorporated into the manuscript at line 240. However, please note that, due to instrumental repairs, we were unable to conduct the release of NPs at different time intervals as requested. We apologize for the inconvenience and regret that we cannot provide the requested data at this time.

Thank you very much for your valuable and insightful comments on improving our articles in a better way. We believe that we have answered all the queries asked by the reviewers and made all the changes in the manuscript as per the Editor and reviewer’s suggestions. We are looking forward to your valuable decision on our revised submission.

Thank you!

  

Round 2

Reviewer 1 Report

Comments and Suggestions for Authors

The author has revised the manuscript according to the reviewers' comments

Reviewer 2 Report

Comments and Suggestions for Authors

The revised version addressed all of my concerns.